# Boosting Lithium Storage of a Metal-Organic Framework via Zinc Doping

**DOI:** 10.3390/ma15124186

**Published:** 2022-06-13

**Authors:** Wenshan Gou, Zhao Xu, Xueyu Lin, Yifei Sun, Xuguang Han, Mengmeng Liu, Yan Zhang

**Affiliations:** 1Institute of Advanced Cross-Field Science, College of Life Sciences, Qingdao University, Qingdao 200671, China; 2019025181@qdu.edu.cn (W.G.); 2019025203@qdu.edu.cn (Z.X.); 2020025520@qdu.edu.cn (Y.S.); 2020025501@qdu.edu.cn (X.H.); 2020025500@qdu.edu.cn (M.L.); 2Beijing National Laboratory for Molecular Sciences and State Key Laboratory of Rare Earth Materials Chemistry and Applications, College of Chemistry and Molecular Engineering, Peking University, Beijing 100871, China

**Keywords:** metal-organic frameworks, zinc-ions doped, energy storage and conversion, lithium-ion batteries

## Abstract

Lithium-ion batteries (LIBs) as a predominant power source are widely used in large-scale energy storage fields. For the next-generation energy storage LIBs, it is primary to seek the high capacity and long lifespan electrode materials. Nickel and purified terephthalic acid-based MOF (Ni-PTA) with a series amounts of zinc dopant (0, 20, 50%) are successfully synthesized in this work and evaluated as anode materials for lithium-ion batteries. Among them, the 20% atom fraction Zn-doped Ni-PTA (Zn_0.2_-Ni-PTA) exhibits a high specific capacity of 921.4 mA h g^−1^ and 739.6 mA h g^−1^ at different current densities of 100 and 500 mA g^−1^ after 100 cycles. The optimized electrochemical performance of Zn_0.2_-Ni-PTA can be attributed to its low charge transfer resistance and high lithium-ion diffusion rate resulting from expanded interplanar spacing after moderate Zn doping. Moreover, a full cell is fabricated based on the LiFePO_4_ cathode and as-prepared MOF. The Zn_0.2_-Ni-PTA shows a reversible specific capacity of 97.9 mA h g^−1^ with 86.1% capacity retention (0.5 C) after 100 cycles, demonstrating the superior electrochemical performance of Zn_0.2_-Ni-PTA anode as a promising candidate for practical lithium-ion batteries.

## 1. Introduction

Lithium-ion batteries (LIBs), due to their high energy density, low cost and long lifespan, have been regarded as critical energy storage devices for electric vehicles, large-scale electricity storage, etc. [1]. The scientific concern with lithium-ion batteries is developing cathodes, anodes, and electrolytes [2,3]. The performance of anode material has become a restriction for high-energy LIBs. As we know, an ideal electrode material should possess both high lithium-ions storage capacity and stable electrochemical performance. The commercial anode material graphite, which can be easily produced, is limited by an insufficient capacity of 372 mA h g^−1^ [4]. Other types of anode materials such as alloy [5,6] and conversion reaction-based transition metal oxides [7,8] possess ultrahigh specific capacity. Nevertheless, the dramatic volume expansion during the charge/discharge process and poor cycle performance restrict the broad use of those materials. Therefore, it is significant to explore novel anode materials with excellent performances for the further development of LIBs.

Metal-organic frameworks (MOFs), as a class of porous materials combining metal ions or clusters with organic linkers through coordination bonds, with a huge variety of structures, large surface areas and adjustable porosity, are widely used in many fields such as gas storage [9], chemical sensors [10], catalysis [11], and drug delivery [12]. Over the past several years, different MOFs have been applied in the secondary battery field, especially in LIBs [13,14]. Generally, the use of MOFs material in the LIBs field can be classified into two aspects, using MOFs as templates to produce homogeneous metal oxide materials and carbon materials for LIBs [14,15,16], and using MOFs as electrode materials directly [17,18,19]. Chen’s group first explored MOF-177 with different morphologies as an anode material for LIBs even though the electrochemical performance of MOF-177 was not promising [18]. After that, more and more MOFs have been investigated as electrode materials for LIBs. For instance, Wang et al. synthesized a Co-based coordination polymer nanowire with a specific capacity of 1132 mA h g^−1^ at a current density of 100 mA g^−1^ [19]. However, most of the works mentioned above focus on introducing novel MOF electrodes for LIBs. Research about the modification of MOFs to improve their inherent property (such as poor electronic conductivity) and electrochemical performances are still scarce [20]. As we all know, element doping is a regular modification to enhance the electronic conductivity and improve the electrochemical performances of electrode materials. Previous similar works could be found in LiFePO_4_ [21,22] and several metal–oxides systems [23]. In this respect, the modification of MOFs by doping should be an effective and practical way to enhance their electrochemical performance. Purified terephthalic acid (PTA) is an optimized ligand used in the synthesis of MOFs due to its easy availability from cheap poly-ethylene terephthalate (PET) plastic products. The purified terephthalic acid-based MOF (Ni-PTA) is a prospective anode material for LIBs due to its unique layered structure. However, the performances of Ni-PTA need to be further elevated by optimizing its fine structure (such as interlayer spacing) to realize the high diffusion of Li^+^ [16].

Compared to Ni^2+^ (0.065 nm), Zn^2+^ owns a larger radius (0.074 nm), and the doping of Zn^2+^ can expand the interplanar distances of Ni-PTA, which facilitates the diffusion of Li^+^. Herein, we successfully synthesized a series of nickel and purified terephthalic acid-based MOF (Ni-PTA) with different amounts of zinc doping (the x% atom fraction Zn-doped Ni-PTA was denoted as Zn_x_-Ni-PTA) through a simple solvothermal method and explored them as anode materials for LIBs. The result shows that Zn_0.2_-Ni-PTA achieves more excellent cycle stability and higher specific capacity than the Ni-PTA without Zn-doped and the 50% atom fraction Zn-doped (Zn_0.5_-Ni-PTA) one. The further kinetics information reveals that the Zn_0.2_-Ni-PTA demonstrates a faster Li^+^ diffusion rate and lower charge transfer resistance, which also explains the brilliant electrochemical performance of Zn_0.2_-Ni-PTA. In addition, we fabricated a full cell based on a LiFePO_4_ cathode to test the electrochemical performance of Zn_0.2_-Ni-PTA as anode.

## 2. Materials and Methods

### 2.1. Synthesis of Zn_x_-Ni-PTA

The Zn_x_-Ni-PTA was prepared by a simple one-pot solvothermal route. Briefly, 0.166 g of purified terephthalic acid (PTA, Aladdin, 99%), 1.5/1.2/0.75 mmol of Ni(NO_3_)_2_·6H_2_O (Aladdin, analytically pure) and different amounts of Zn(NO_3_)_2_·6H_2_O (Aladdin, analytically pure) (0, 0.3, 0.75 mmol) were, respectively, dissolved in a mixed solvent consisting of 15 mL absolute ethanol (Macklin, analytically pure) and 15 mL deionized H_2_O, which was then stirred for 30 min. Then, the mixture was transformed into a 40 mL Teflon-lined stainless steel autoclave and reacted at 180 °C for 24 h. After the autoclave cooled down to room temperature, the green precipitates were washed by N,N-dimethylformamide (DMF, Aladdin, analytically pure) and absolute ethanol several times. Ultimately, this product was dried at 60 °C in air for 24 h. The 0%, 20% and 50% atom fraction Zn-doped Ni-PTA denoted Zn_0_-Ni-PTA, Zn_0.2_-Ni-PTA, and Zn_0.5_-Ni-PTA, respectively.

### 2.2. Materials Characterization

The powder X-ray diffraction (XRD) patterns were recorded by a Rigaku Ⅱ X-ray diffraction spectrometer (Japan Science Co., Tokyo, Japan) using Cu-Kα radiation. Fourier-transform infrared (FTIR) transmission spectra were performed by FTIR-65 IR spectrophotometer (Tianjin Port East Technology Co., Tianjin, China). Thermogravimetric–Differential Thermal Analysis (TG-DTA) was performed by a Labsys Evo thermogravimetric differential thermal analyzer from the Setaram Instrumentation (France) with a rate of 10 K min^−1^ under ambient condition. The JSM-JSM7500 instrument (Japan Electronics Co., Tokyo, Japan) obtained scanning electron microscopy (SEM) images. X-ray photoelectron spectroscopy (XPS) was taken on the PHI5000VersaProbe instrument (Shanghai Yuzhong Industrial Co., Shanghai, China). Inductively Coupled Plasma–Atomic Emission Spectrometry (ICP-AES) record was obtained by ICP-9000(N+M) atomic emission spectroscopy (Thermo Jarrel-Ash Co., Boston, MA, USA).

### 2.3. Electrochemical Measurements

To carry out the electrochemical measurements, the active material, Super-P carbon black and polyvinyl difluoride (PVDF, Macklin, *M*_w_ = 1,000,000) were blended in a weight ratio of 6:3:1 with several drops of N-methyl pyrrolidone (NMP) added and stirred until the mixture became homogeneous. Afterward, the mixed slurry was pasted onto a copper foil current collector with a diameter of 10 mm and dried in vacuum at 80 °C for 12 h. The average loading of active materials was about 1.2 mg cm^−1^. The CR 2016-type cells were assembled in an Ar-filled glove box (water and oxygen concentration less than 0.1 ppm), using lithium foil as the counter electrode and Celgard 2300 polypropylene as separators (diameter—16 mm). The electrolyte was 1 M LiPF_6_ dissolved in a solvent mixed with EC-DMC-EMC (1:1:1 vol %). The galvanostatic charge–discharge (GCD) tests were performed by a LAND CT2001 battery test system (Wuhan Kingnuo Electronic Co., Wuhan, China) at room temperature (25 °C). Cycle voltammetry (CV) and electrochemical impedance spectroscopy (EIS) were taken on a CHI-660B electrochemical station (Shanghai Chenhua Instrument Co., Shanghai, China) at a full charge state of batteries after the 20th cycle. The current density of test is 0.1 A g^−1^. The CV tests were carried out with a scan rate of 0.1 mV s^−1^, while the EIS data were recorded in the frequency range of 0.01–100 kHz. The full cell was similarly assembled as above-mentioned: 20 at % Zn-doped Ni-PTA as anode, and the cathode was obtained by homogeneously mixing the LiFePO_4_ (Canrd, D-1, 80 wt %), PVDF (10 wt %) and NMP (10 wt %). The loading density was about 0.4 mg cm^−2^, and Al foil was used as the current collector. The specific capacity was calculated based on the cathode material.

## 3. Results and Discussion

### 3.1. Characterization of Zn_x_-Ni-PTA

To investigate the crystalline phase of as-prepared products, XRD measurement was performed. Figure 1a shows the XRD patterns of Zn_x_-Ni-PTA with different amounts of zinc doping and the standard card of Ni_3_(OH)_2_(C_8_H_4_O_4_)_2_·(H_2_O)_4_·2H_2_O (CCDC 638866) belonging to the space group of P-1(2)-triclinic. Although all the XRD patterns of synthesized MOFs shared similarities, with the increasing Zn^2+^ amount, the peaks of (010) and (020) tend to become invisible. This phenomenon illustrates that the crystal structure order along the *b*-axis decreased. In addition to the absence of some reflections, with the amount of zinc dopant increasing, the peaks of (100) and (200) shifted to a lower angle, which can be attributed to the partial replacement of doped larger Zn^2+^ (0.074 nm) to Ni^2+^ (0.065 nm) in MOFs, expanding the interplanar distances along the *a*-axis [24,25]. The SEM images of Zn_x_-Ni-PTA are presented in Appendix A. The Zn_0_-Ni-PTA showed layered micro-sheet like morphology features, and the flower-like layered structure was observed after Zn^2+^ doping. TGA curves (Appendix A) indicates that the initial weight loss is due to the loss of solvated water molecules and the weight loss in the range of 350–420 °C is attributed to the thermal decomposition of the Zn_x_-Ni-PTA.

In addition, the similar FTIR peaks also proved that these synthesized MOFs have similar layered topology crystal structures, as shown in Figure 2a. According to the patterns, bands at 1501 cm^−1^ reveal the para-aromatic C-H stretching mode. The bending stretching vibration of -OH is observed at 3613 cm^−1^. The bands at 3070 cm^−1^, 3344 cm^−1^ and 3431 cm^−1^ are assigned to stretching vibrations of H_2_O in MOFs. In addition, the asymmetric and symmetric vibration of COO^−^ are located at 1581 cm^−1^ and 1400 cm^−1^, respectively. The more Zn content, the larger of COO^−^ groups separation that demonstrated the impact of doped zinc ions on the structure of MOF. The absorption peaks at 522 cm^−1^ in Zn_0,0.2,0.5_-Ni-PTA are associated to Ni-O, and peaks at 437 cm^−1^ in Zn_0.2,0.5_-Ni-PTA are associated to Zn-O vibration bonds, respectively [26,27]. This also confirms the successful doping of Zn^2+^ to the Ni-PTA. The content of dopant Zn^2+^ in Zn_x_-Ni-PTA samples (0%, 22.06% and 50.83%) was detected by induced coupled plasma atomic emission spectroscopy (ICP-AES), respectively (Appendix A) and matched well with the theoretical value. The above results indicated that different amounts of Zn^2+^ successfully replaced the Ni^2+^ in Ni-MOFs.

For further information about elemental composition and valance state, an XPS test was performed. As observed from the XPS spectra of Zn 2p (Figure 2b), no characteristic peaks were detected in the Zn_0_-Ni-PTA sample, and two prominent doublet peaks corresponding to Zn 2p_1/2_ and Zn 2p_3/2_ can be observed in Zn_0.2_-Ni-PTA and Zn_0.5_-Ni-PTA, indicating the successful doping of Zn in these two samples [28]. In the Ni 2p spectra (Figure 2c) of Zn_x_-Ni-PTA, all samples share similar peak types attributed to Ni^2+^, but the binding energy shows the obvious difference before and after doping. The binding energy of Ni 2p_3/2_ in pristine Ni-PTA (Zn_0_-Ni-PTA) is 855.8 eV; by contrast, the value shifts to 858.2 and 857.9 eV in Zn_0.2_-Ni-PTA and Zn_0.5_-Ni-PTA. Such difference perhaps originates from that the doped Zn^2+^ with different electronegativity influences the electronic interactions of Ni^2+^ [29,30]. Based on the discussion above, it can be inferred that Zn^2+^ has strong bands with organic ligands in MOFs. All of the evidence discussed above proved the successful doping of zinc. The surface area (BET) date of different Zn_x_-Ni-MOF are display in Appendix A.

### 3.2. Half-Cell Test

To investigate the Li-ion storage capacity of Zn_0.2_-Ni-PTA, electrochemical tests were performed and are shown in Figure 3. The CV curves, as shown in Figure 3a, exhibit a weak reduction peak at about 1.30 V and a sharp reduction peak at 0.75 V followed by a broad oxidation peak around 1.25 V during the initial anodic scan, which can be ascribed to the lithiation/de-lithiation of carboxylate groups and aromatic rings and the formation of a solid-electrolyte interphase (SEI) layer [31,32]. Compared with the first cycle, the intensity of the redox peaks in the second and third cycles are growing weaker, which is related to structural or morphological evolution [33]. The sharp redox peaks shift toward higher/lower potential (≈0.75/1.40 V), which corresponds to the Li-ions insertion/extraction into/from the carboxylate groups and the benzene rings of Zn_x_-Ni-PTA [34]. Notably, Zn_0_-Ni-PTA, Zn_0.2_-Ni-PTA, and Zn_0.5_-Ni-PTA showed similar CV curves (Appendix A left). This phenomenon implies that zinc dopant did not change the topology of pristine MOF; instead, it only just boosted lithium storage performance. The charge–discharge curves of the Zn_0.2_-Ni-PTA are shown in Figure 3b. Two discharge voltage plateaus at about 1.30 V and 0.75 V could be observed in the first discharge curve. Then, following the charging process, two charge ranges are demonstrated, which agree well with the CV curves. Combining with the subsequent cycles, we can assume that the reversible discharge plateau of Zn_0.2_-Ni-PTA at 1.0–0.5 V contributes the most capacity.

The cycle performance of different amounts of Zn-doped Ni-PTA at the current density 0.1 A g^−1^ and 0.5 A g^−1^ is displayed in Figure 3c,e. The 20% atom fraction Zn-doped Ni-PTA delivers the initial discharge capacity of 2000 mA h g^−1^ at 0.1 A g^−1^ and 1690 mA h g^−1^ at 0.5 A g^−1^. After 100/200 cycles, the discharge capacity retains up to 921 and 747 mA h g^−1^ at 0.1 A g^−1^/0.5 A g^−1^, respectively, with a Coulombic efficiency of nearly 100%. We are noticing that the capacity of three samples mildly increases after about the 10th cycle, which might be due to the electrochemical activation process related to the repeated insertion/extraction of Li-ions in MOFs. In simple terms, during charging and discharging, the inserted Li^+^ expanded the interlayer spacing of samples and exposed a larger number of active sites, which could be beneficial for gradually increasing capacity. In addition, it is clear that the cycle performance of Zn_0.2_-Ni-PTA is superior to the other two MOFs. In addition, the better rate performance of Zn_0.2_-Ni-PTA is exhibited in Figure 3d. In summary, the reversible capacity at current density values of 50, 100, 300, and 500 mA g^−1^ is 958, 806, 646, and 556 mA h g^−1^, respectively. Moreover, when the current density decreases to 50 mA g^−1^, the discharge capacity remains stable.

Based on the above electrochemical tests, combined with the cycle performance information of different atom fractions of (0, 10, 20, 30, 40, 50%) Zn-doped Ni-PTA (Appendix A), we can prove that Zn_0.2_-Ni-PTA displays the best electrochemical performance among different atom fractions of zinc doped Ni-PTA samples. For comparison, their electrochemical performances are presented along with the other MOFs in Appendix A. In addition, the XRD patterns and SEM images of Zn_0.2_-Ni-PTA electrodes before and after the cycle are displayed in Appendix A. The SEM images show that Zn_0.2_-Ni-PTA maintains its pristine microplate-like characteristics after cycles, indicating its good morphological stability. Interestingly, an amorphization process after cycles is determined by XRD. This phenomenon perhaps originates from the electrochemical powderization effect due to the repeat insertion/extraction of Li^+^ in its crystal [35]. Such an amorphization process was also observed in other lithium storage cases of MOFs materials [33,35]. Moreover, the decrease in redox peaks in the first few cycles observed from CV curves may be also related to this amorphization process.

### 3.3. Electrochemical Analysis

To better understand why Zn_0.2_-Ni-PTA possesses brilliant electrochemical performance, the effect of doped Zn^2+^ on the migration of Li^+^ was also investigated. It is well known that the high migration rate of lithium-ions can improve the electrochemical performance of the battery [36,37,38]. Therefore, electrochemical impedance spectroscopy (EIS) experiments are conducted to explore the kinetics. From the Nyquist diagrams, as shown in Figure 4a, it is shown that Zn_0.2_-Ni-PTA possesses a lower electrochemical impedance than Zn_0_-Ni-PTA. Furthermore, the chemical diffusion coefficient of Li^+^ (*D*_Li+_) was calculated by Equation (1). R, *T*, *A*, *F*, *n*, and *C* are the ideal gas constant, the thermal–dynamic temperature, the surface area of electrodes, the Faraday’s constant, the number of electrons per molecule during oxidation, and the Li^+^ concentration in the cathode, respectively. The *σ_W_* is the Warburg coefficient, which has a linear relation with *Z’* (Equation (2)):(1)$$D_{{\mathrm{Li}}^{+}}={\left(\frac{2RT}{\sqrt{2}n^{2}F^{2}\sigma_{W}AC}\right)}^{2}=\frac{2R^{2}T^{2}}{n^{4}F^{4}\sigma_{W}^{2}A^{2}C^{2}}$$
(2)$$Z^{\prime }=R_{s}+R_{ct}+\sigma_{W}\omega^{-1/2}$$

In Equation (2), *R_s_* and *R_ct_* represent the resistance of solution and the charge transfer, respectively [25]. Figure 4b illustrates the linear relationship plot of *ω*^−1/2^ (reciprocal square root of angular frequency) vs. *Z’* (actual impedance) in the low frequency of Zn_x_-Ni-PTA. The slope of this linear plot is equal to the value of σ_W_. As shown clearly in the plot, by contrast with Zn_0_-Ni-PTA, Zn_0.2_-Ni-PTA possess a smaller slope, which demonstrates a higher Li^+^ diffusion coefficient and better electrochemical performance. Compared to Ni^2+^ (0.065 nm), Zn^2+^ has a larger radius (0.074 nm), and the doped Zn^2+^ expands the interplanar distances of Zn_0.2_-Ni-PTA, which facilitates the diffusion of Li^+^. Thus, Zn_0.2_-Ni-PTA displays a higher Li^+^ diffusion coefficient and improved lithium storage performances [39].

### 3.4. Full-Cell Test

To further investigate the application prospects of the sample, we constructed the full cell using Zn_x_-Ni-PTA as anode coupled with LiFePO_4_ as cathode. The electrochemical test is performed at a voltage range of 0.5-4.0 V. Figure 5a shows the charge and discharge curves of the Zn_0.2_-Ni-PTA/LiFePO_4_ full cell. It yields a discharge capacity of 113.7 mA h g^−1^ with an average operation voltage of around 2.6 V at 0.5 C. At 0.1 C, 0.5 C, 1 C, 2 C, and 5 C, the reversible capacity remains 124.5, 105.9, 82.2, 62.4, and 50.3 mA h g^−1^, respectively (Figure 5b). After 200 cycles, the capacity still retains 94.7 mA h g^−1^ with a high capacity retention of 83.24%, which contrasted sharply with the terrible cycle performance of Ni-PTA without Zn doped (Figure 5c). It should be mentioned that the mass loading of LiFePO_4_ was relatively lower in this work compared to that in commercial LIBs. For most lab-level research, the mass loading of active materials is also lower [40]. In consideration of practical application, we will try modifying the loading of LFP in our later research to achieve a higher energy density.

The above electrochemical result of both half and full cells based on Zn_0.2_-Ni-PTA with high reversible capacity, safe operation potential, excellent rate and cycle stability is superior to the previous report of synthesized Ni-MOF [16]. This could be attributed to the following aspects: First, the zinc with a large radius can hinder the destruction of the electrode material and enlarge the interplanar distances [21,24,31,32]. Meanwhile, the EIS results showed that Zn_0.2_-Ni-PTA delivered less impedance and charge-transfer resistance, bringing a faster Li-ion transmission rate. In summary, Zn_0.2_-Ni-PTA is a valuable anode material for LIBs.

## 4. Conclusions

In this work, we successfully synthesized different amounts of Zn doped Ni-PTA and used it as an electrode material for LIBs. In contrast with Ni-PTA, the visible enhancement of lithium storage ability (initial discharge capacity is 2000 mA h g^−1^ at the current density of 100 mA g^−1^) and cycle stability (≈85% reversible discharge capacity retained after 100 cycles) are observed in 20% Zn-doped Ni-PTA. The following characterizations indicated that the better performance of samples is because of the higher structural stability and lower impedance and charge-transfer resistance after zinc ions are doped. The work brings new perspectives to the modification of present MOFs electrode materials for LIBs. It may lead us to think about the potential use of Zn_0.2_-Ni-PTA as anode materials for lithium-ion batteries in the future.

## Figures and Tables

**Figure 1 materials-15-04186-f001:**
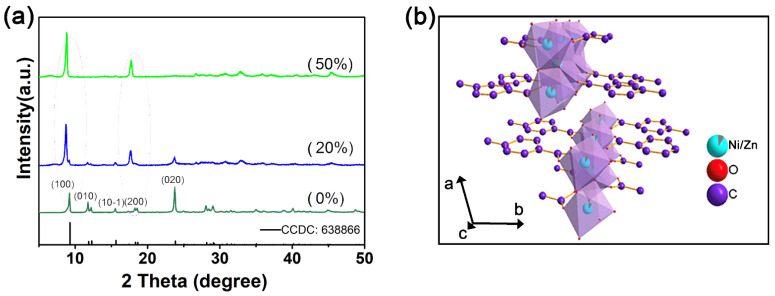
(**a**) Powder XRD pattern of the different atom fraction Zn_x_-Ni-PTA samples. (**b**) The structure mode of Zn_x_-Ni-PTA.

**Figure 2 materials-15-04186-f002:**
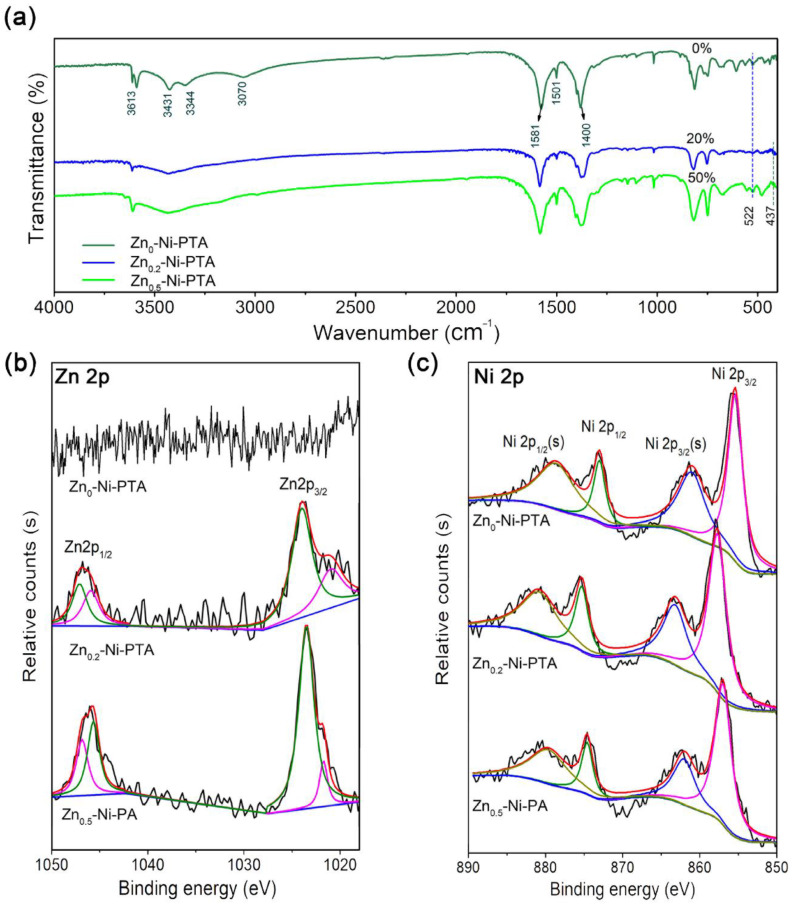
(**a**) FTIR pattern of the different atom fraction Zn_x_-Ni-PTA samples. (**b**) Zn 2p and (**c**) Ni 2p spectra of Zn_x_-Ni-PTA.

**Figure 3 materials-15-04186-f003:**
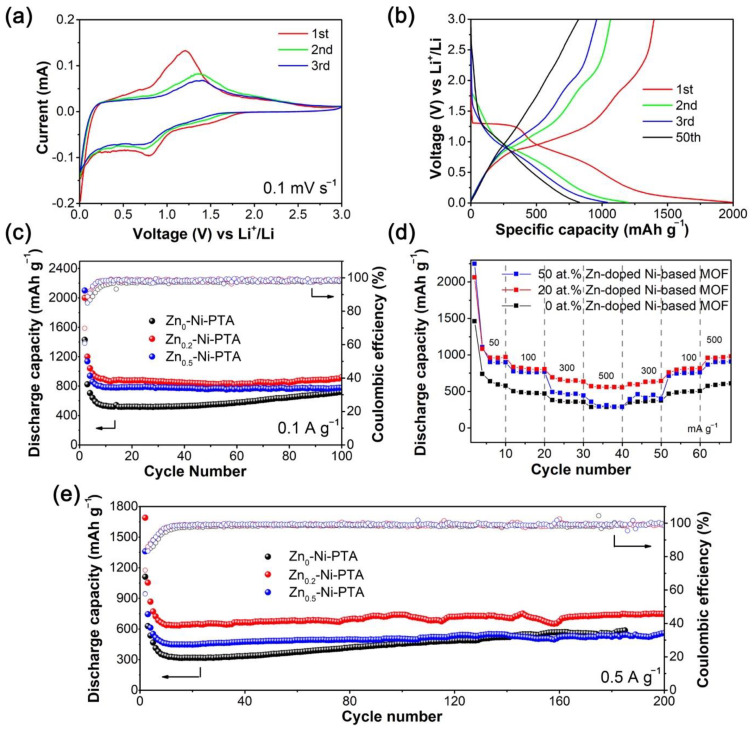
Electrochemical performance of Zn_x_-Ni-PTA material: (**a**) CV curves and (**b**) voltage profiles of 20% atom fraction Zn-doped Ni-PTA. (**c**,**e**) Cycle and (**d**) rate performance of different MOFs.

**Figure 4 materials-15-04186-f004:**
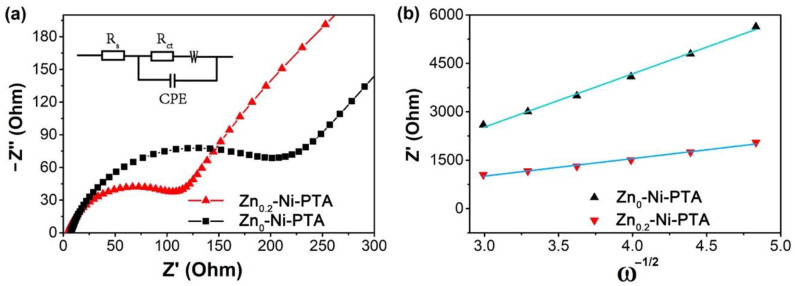
(**a**) Nyquist plot of Zn_0.2_-Ni-PTA and Zn_0_-Ni-PTA. (**b**) Z’ vs. ω^−1/2^ plots in low frequency region.

**Figure 5 materials-15-04186-f005:**
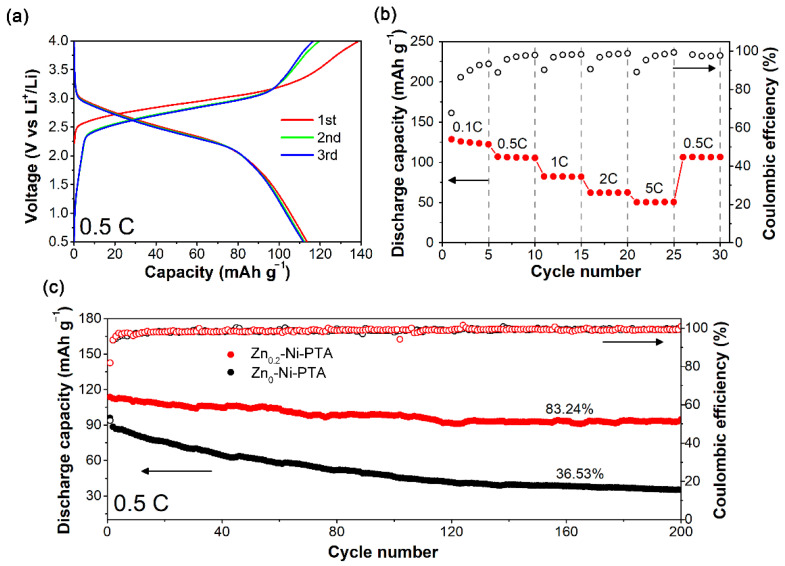
(**a**) Charge–discharge curves, (**b**) rate performance and (**c**) cycle performance of Zn_0.2_-Ni-PTA/LiFePO_4_ full cell.

## Data Availability

Not applicable.

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
