# Peer review of "Boosting Lithium Storage of a Metal-Organic Framework via Zinc Doping"

_materials, 2022, doi:10.3390/ma15124186_

Round 1

Reviewer 1 Report

In this article nickel and terephthalic acid-based MOF has been doped with different amounts of Zn and applied as electrode material for lithium storage in lithium ion batteries. The Zn doped MOFs have been characterized with XRD, FTIR, SEM and XPS analysis. It was found that 20% atom fraction Zn doped Ni-PTA (Zn0.2-Ni-PTA) boosted lithium storage capacity to a great extent. The work is interesting and applied nature. The article is a good written scientific report. However, there are several shortcomings in the manuscript which must be addressed before its publication in Materials. My comments are as under

  1. Revise the sentence “Other broadly used anode materials such as alloy type anodes [5, 6]; conversion reaction-based transition metal oxides [7, 8] exhibit ultrahigh specific capacity but are still restricted by poor cycle performance resulting from the dramatic volume expansion during the charge/discharge process” in the introduction.
  2. In the introduction the selection of Ni-PTA as MOF has not been justified. Similarly the selection of Zn as dopant needs further reasoning. It is suggested to revise the introduction to highlight novelty by writing comprehensive problem statement.
  3. XRD patterns are described very superficially. What are the fates of other peaks (other than 100) such as 200 in Fig.1 spectra b and c.
  4. On cannot see any band in the FTIR spectra for the presence of Ni and Zn in the MOF.
  5. Do the authors have a valid reason for the decrease in intensity of the redox peaks in 2nd and 3rd cycle as compared to first cycle in fig.3a?
  6. They write " We are noticing that the capacity of three samples gradually increases cycling, which might owe  to the electrochemical activation process related to repeat insertion/extraction of Li-ions in MOFs". this explanation simply nullifies the observation of decrease in redox peaks with cycling shown in figure 3a.
  7. There are several typo/grammatical mistakes in the text.

Author Response

Thanks for your kind and professional comments on our work and they are beneficial for us to improve our manuscript. A point-by-point response is provided in the uploaded file.

Reviewer 2 Report

The work by Gou et al. successfully synthesize nickel and purified terephthalic acid-based MOF (Ni-PTA) with a series amounts of zinc dopant (0, 20, 50%) and evaluate them as anode materials for lithium-ion batteries. It shows that the 20% atom fraction Zn-doped Ni-PTA (Zn0.2-Ni-PTA) exhibits a high specific capacity of 921.4 mA h g-1 and 739.6 mA h g-1 at different current density of 100, 500 mA g-1 after 100 cycles.

The manuscript is overall well organized with all the results clearly presented and discussed. The scope also aligns well with the selected topic of Materials. The reviewer would be happy to recommend its acceptance of publication in Materials after necessary revisions.

1. The loading of LFP is only 0.4 mg/cm2, which is too low for full-cell demonstration, making the total capacity be of little practical value. The authors need to clarify this point as to strengthen the conclusion from the full-cell results.

2. At which state were the EIS spectra acquired for diffusivity calculation? 

3. What are the redox reactions corresponding to the 1.3 and 0.75 V plateaus shown in Fig. 3? The authors need to at least discuss the electrochemical reaction mechanisms for future reference. 

4. How about the morphologies after cycling for different electrodes? Was the 20% Zn-doped one also showed advantage in inhibit volume expansion/cracking.

Author Response

Thanks for your kind and professional comments on our work and they are beneficial for us to improve our manuscript. A point-by-point response has been provided in uploaded file.

Round 2

Reviewer 1 Report

Though the authors have revised the article to a satisfactory level. Most of my earlier comments have been complied. It may be published in materials after thorough checking for English.